# Development and Determinants of Topsoil Bacterial and Fungal Communities of Afforestation by Aerial Sowing in Tengger Desert, China

**DOI:** 10.3390/jof9040399

**Published:** 2023-03-24

**Authors:** Weiyu Chen, Tengfei Yu, Chenguang Zhao, Baofeng Li, Yanyan Qin, Huiying Li, Haojie Tang, Junliang Liu, Xiaoyou Zhang

**Affiliations:** 1Key Laboratory of Ecological Safety and Sustainable Development in Arid Lands, Northwest Institute of Eco-Environment and Resources, Chinese Academy of Sciences, Lanzhou 730000, China; 2Alxa Desert Eco-Hydrology Experimental Research Station, Northwest Institute of Eco-Environment and Resources, Chinese Academy of Sciences, Lanzhou 730000, China; 3University of Chinese Academy of Sciences, Beijing 100049, China; 4Alxa Institute of Forestry and Grassland, Alxa 750306, China; 5Alxa Forestry and Grassland Projection Station, Alxa 750306, China

**Keywords:** soil microbe, afforestation, diversity, community composition, desert

## Abstract

It was previously reported that afforestation in the desert can help improve soil texture, carbon accumulation, and nutrient status. However, the effects of afforestation on soil microbial composition, diversity, and microbial interactions with soil physicochemical properties have been rarely evaluated quantitatively. Using the method of space-for-time substitutions, we assessed the development and determinants of topsoil bacterial and fungal communities over nearly 40 years of successive afforestation by aerial sowing in Tengger Desert, China. The results showed that afforestation by aerial sowing comprised a considerable proportion of Chloroflexi and Acidobacteria in the bacterial community in addition to the ubiquitous phyla found in desert but had fewer effects on the dominant phyla of the fungal community. At the phylum level, the bacterial community was clearly clustered into two groups. However, it was difficult to differentiate the constituents of the fungal community based on principal coordinate analysis. The richness of the bacterial and fungal communities was significantly higher after five years than at zero years and three years. Additionally, the bacterial community varied parabolically and reached its largest size at twenty years, while the fungal community increased exponentially. Soil physicochemical properties were found to have divergent effects on the abundance and diversity of bacterial and fungal communities, among which salt- and carbon-associated properties (e.g., electrical conductivity, calcium, magnesium, total carbon, and organic carbon) were closely related with the abundance of bacterial-dominant phyla and the diversity of bacteria and fungi, but nutrient-associated properties (e.g., total phosphorus and available phosphorus) were not. The results indicate that afforestation through the salt secretions of plants leaves and carbon inputs from litter promote the development of topsoil bacterial and fungal communities in the desert.

## 1. Introduction

Deserts, which constitute one fifth of the global terrestrial surface, are characterized by extreme fluctuations in temperature, low water and nutrient availability, high ultraviolet solar radiation, and intense winds [1]. In China, the desert and desertification-prone region (DPR) are presently 4.21 × 10^6^ km^2^ in size, occupying 44% of China’s land area; these regions confront the great challenges in combating desertification [2]. Because of the extreme and reduced-complexity environments of deserts, soil microbiomes are likely the dominant drivers in these systems. Thus, deserts offer a tremendous opportunity for studying the structure, function, and evolution of natural microbial communities [3,4]. Cross-biome metagenomic analyses showed that desert biomes are markedly distinct from other biomes in terms of the soil microbial community composition and diversity [1,5]. 

Large-scale afforestation in the desert and DPR are presently one of the most effective approaches to combat desertification [6]. After afforestation, large amounts of silt- and clay-sized particles are deposited onto the dune surfaces, altering the composition of the original soil texture [7,8,9]. Then, soil salt and nutrients are excreted on the top soil via plant litter [10,11], thereby shifting the composition and diversity of the microbiome [12,13,14]. In addition to a number of ubiquitous bacterial phyla including Actinobacteria, Proteobacteria, Bacteroidetes, Gemmatimonadetes, and Firmicutes [1], as well as fungal phyla including Ascomycota and Basidiomycota [15], some specific bacterial phyla (e.g., Chloroflexi and Acidobacteria) have also been found in the desert soil after afforestation [12,16,17]. However, the prevalence of such phyla must still be studied in more detail. Moreover, afforestation on shifting sand dunes has significantly increased the diversity and richness of bacterial communities with *Haloxylon ammodendron* (C. A. Mey.) Bunge planted in the arid desert [12] and fungal communities after planted with *Caragana microphylla* Lam. in the semi-arid desert [17], suggesting that the effect of afforestation on microbial diversity has an uncertain association with species, climates, and soil properties.

It is well-known that soil microbial communities are strongly influenced by soil physicochemical properties, which could explain a large proportion of the variance in soil microbial community composition and diversity at different spatial scales [18,19,20]. However, the determinants of microbial community composition and diversity differ between sites with different species [12,16,17,21,22]. For example, total carbon, pH, and total phosphorus were found to be major abiotic factors driving the structures of bacterial communities under artificial revegetation in an arid oasis–desert ecotone [12]. Since the late 1970s, China has taken many measures to slow desertification, including fencing of natural vegetation and afforestation through tree planting and aerial sowing [23]. Afforestation by aerial sowing has been used at a large scale, as this technique can quickly cover a large area and reach physically inaccessible areas [24,25]. In contrast with afforestation by tree planting in the desert [12,16,17,21], little is known about the specific mechanisms of microbial survival and adaption under afforestation by aerial sowing, which greatly limits our understanding of the vegetation community succession of this ecosystem. 

In the study, we aimed to characterize the development of the composition and diversity of soil microbial communities and identify the determinants that shape soil microbial communities in the desert using nearly 40 years of succession following afforestation by aerial sowing in the northeastern area of Tengger Desert, China. Specifically, we hypothesized that (1) the composition of the microbial community would change with plant community succession, (2) the diversity of the microbial community would increase with an increase in the stage of afforestation, and (3) the soil salt and nutrient status would shape the composition and determine the diversity of the microbial community.

## 2. Materials and Methods

### 2.1. Study Sites and Sample Collection

The study sites were located at the northeastern edge of Tengger Desert, China (Appendix A). The sites had cold-winter-desert climates according to the Köppen–Geiger climate classification [26]. The mean annual precipitation was 145 mm, of which 62% was concentrated in June to September. The mean annual temperature was 8.4 °C, with an historical extreme minimum of −34.4 °C in January and a maximum of 41.1 °C in July. The mean annual wind speed was 7.1 m·s^−1^, and the maximum reached 26 m·s^−1^ in spring and winter. The area suffered from strong wind erosion year-round, so the surface mainly consisted of fixed and semi-fixed dune landforms. The soil was dominated by non-zonal aeolian soil, and the texture was sandy. Since 1984, afforestation by aerial sowing has been carried out in the bare desert with selected shrub species including *Hedysarum scoparium*, *Calligonum mongolicum* Turcz., and *A. sieversiana* Ehrhart ex Willd.

The field investigation was carried out in the peak growing season from June to July in 2021. Using the method of space-for-time substitutions, which can be reliably applied to study aspects of soil development at time scales ranging from centuries to millennia [27], we selected seven stages with aerial sowing in 2021 (A0), 2018 (A3), 2016 (A5), 2010 (A11), 2001 (A20), 1992 (A29), and 1984 (A37) to establish a nearly 40-year sequence of vegetation succession (Appendix A). We established five repeated sampling sites 30 × 30 m in size from each selected stage at an interval of 2–3 km with a total of 35 sampling sites. Three 10 × 10 m sampling plots were set up in each sampling site along the southwest to northeast diagonal, which were the basic units for the investigation of plant community characteristics and soil sampling (Appendix A). Three soil samples were taken from 0–10 cm and 10–20 cm, respectively, mixed into a composite sample and used to the DNA extraction and sequencing. Meanwhile, three fresh soil samples about 500 g in weight were collected and mixed to analyze the soil physicochemical properties. 

### 2.2. Soil Physicochemical Properties

Soil pH and electrical conductivity (EC, μS/cm) were measured using the glass electrode method (FE28, Mettler Toledo, Greifensee, Switzerland) with a 1:2.5 and 1:5 ratio of soil mass to water [17]. Total carbon (TC, g/kg) and nitrogen (TN, g/kg) were determined using a C/N element analyzer (CN802, VELP, Usmate Velate, Italy). Soil organic carbon (SOC, g/kg) was measured using the potassium permanganate method with external heating. Total phosphorus (TP, g/kg) was determined using a digester (HYP-320, Shanghai, China) and flow injector (Foss Fiaster 5000, Höganäs, Sweden). Available potassium (AK, mg/kg) and phosphorus (AP, mg/kg) were determined via exchangeable cation extraction in an ion-exchange resin and then measured using flame photometry and photoelectric colorimetry, respectively. Soil cation concentrations (mg/L), including sodium (Na), potassium (K), calcium (Ca), and magnesium (Mg), were measured using an ion chromatography system (ICS-2500, Dionex Cor., Sunnyvale, CA, USA). Aside from pH, the afforestation stages had significant effects on soil physicochemical properties (Appendix A).

### 2.3. DNA Extraction and Sequencing

Total microbial genomic DNA was extracted from 70 samples (35 sites × 2 depths) using an E.Z.N.A.^®^ Soil DNA Kit (Omega Bio-tek, Norcross, GA, USA). The quality and concentration of DNA were determined by 1.0% agarose gel electrophoresis and a NanoDrop^®^ ND-2000 (Thermo Scientific Inc., Waltham, MA, USA), respectively. The 16S rRNA gene was amplified by PCR using the primer pairs 338F (5′-ACTCCTACGGGAGGCAGCAG-3′) and 806R (5′-GGACTACHVGGGTWTCTAAT-3′). The ITS rDNA gene was amplified by PCR using the primer pairs ITS1F (CTTGGTCATTTAGAGGAAGTAA) and ITS2R (GCTGCGTTCTTCATCGATGC) with an ABI GeneAmp^®^ 9700 PCR thermocycler (ABI, Los Angeles, CA, USA). The 20 µL PCR reaction mixture consisted of 10 ng template DNA and ddH_2_O, 4 μL 5× Fast Pfu buffer, 2 μL 2.5 mM dNTPs, 0.8 μL of each primer (5 μM), 0.4 μL Fast Pfu polymerase, and 0.2 μL BSA. PCR amplification conditions were designed following the Shanghai Majorbio Bio-Pharm Technology Co., Ltd. (Shanghai, China). All samples were amplified in triplicate. Agarose gel with a 2% concentration was used to extract the PCR product, and an AxyPrep DNA Gel Extraction Kit (Axygen Biosciences, Union City, CA, USA) was employed to purify it. Purified amplicons were blended in equimolar amounts and paired-end sequenced on an Illumina MiSeq PE300 platform (Illumina, San Diego, CA, USA). 

### 2.4. Data Processing

Raw sequencing data were de-multiplexed using an in-house Perl script and then quality-filtered with FASTP software (version 0.19.6) [28] and merged using FLASH software (version 1.2.7) [29]. The operational taxonomic units (OTUs) were obtained by clustering the optimized sequences at a 97% similarity level using UPARSE software [30]. A representative sequence for each OTU was selected as the most abundant sequence. Then, the sequence was annotated and classified based on the RDP classifier Bayesian algorithm for bacteria and the Unite database (http://unite.ut.ee/index.php (accessed on 13 September 2021)) for fungi. To minimize the effects of sequencing depth on microbial diversity, each OTU was subsampled according to the lowest number of sequences in all samples. The rarefaction curves at the OTU level showed clear asymptotes (Appendix A), and the Good’s coverages were 97.19 ± 0.10% and 99.88 ± 0.01% for bacteria and fungi, respectively, which together indicated nearly complete sampling of the community. 

### 2.5. Bioinformatic Analysis

Bioinformatic analysis of the soil microbiota was carried out using the Majorbio Cloud Platform (http://www.majorbio.com (accessed on 28 July 2022)). Community composition was visualized using a bar diagram. A Kruskal–Wallis H test of multiple comparisons (*p* < 0.05) was performed to detect the significance of the differences in community composition between stages using the R Vegan package (v2.6-4). Alpha (*α*-) diversity, including richness indices of the observed species (Sobs), ACE, and Chao1 and diversity indices of Shannon, Simpson, and Faith’s phylogenetic diversity (PD) at the OTU level, were obtained using Mothur (version 1.30.2). Principal coordinate analysis (PCoA) based on Bray–Curtis distance was used to visualize the variation in the community structure between stages, and the intra-group differences were tested using Adonis with 999 permutations at the OTUs level.

### 2.6. Determinants of Microbial Communities 

A Spearman correlation heatmap was used to visualize the relationship between the abundance of dominant phyla and soil physicochemical properties using the R pheatmap package [31]. Redundancy analysis was performed to investigate the determinants of abundance in the microbial community in terms of soil physicochemical properties using the R vegan package. Mental’s test was employed to determine the relationship between diversity and soil physicochemical properties by calculating the correlation matrix with Euclidean distances using the R linkET package. A Random Forest was used to quantify the relative contribution of soil physicochemical properties to the *α*-diversity of the microbial community. The RF parameters were set to 500 decision trees and 5 predictors sampled for splitting at each node. Next, 99 permutations were performed to test the significance of the global model under a level of *p* = 0.05, and then the significance of each critical metric for the model was determined by permuting the response variable using the rfPermute package [32].

## 3. Results

### 3.1. Soil Microbial Community Composition

For bacteria, a total of 1,996,260 high-quality sequences were obtained and clustered into 11,790 OTUs across 70 soil samples. The 2424 classified species were derived from 45 phyla, 144 classes, 366 orders, 607 families, and 1190 genera. The dominant phyla (top 10) in order were Actinobacteria, Proteobacteria, Chloroflexi, Acidobacteria, Bacteroidota, Gemmatimonadota, Firmicutes, Myxococcota, Patescibacteria, and Planctomycetota, which together account for >95% of the bacterial community in different stages (Figure 1); additionally, the phyla can be hierarchically clustered into one group except for the last two phyla (Appendix A). The composition of the dominant phyla differed significantly between stages (*p* < 0.05), except for Proteobacteria, Bacteroidota, and Firmicutes at depths of 0–10 cm and Firmicutes and Myxococcota at depths of 10–20 cm (Figure 1A,B). Particularly, the second-most dominant phyla, Chloroflexi and Acidobacteria, accounted for a higher proportion, with 4.3–14.2% and 1.9–20.6%, respectively, and varied drastically between stages. Composition of the dominant phyla (top 10) remained unchanged at the two depths, but the proportion of Bacteroidota was less than that of Gemmatimonadota and Firmicutes at depths of 10–20 cm (Figure 1B). 

A total of 2,221,800 high-quality sequences from the fungal community were obtained and clustered into 2929 OTUs across 70 soil samples. The 1065 classified species were from 12 phyla, 40 classes, 102 orders, 246 families, and 520 genera. Except for unclassified taxa, the dominant phyla (top 5) were Ascomycota, Basidiomycota, Mortierellomycota, Glomeromycota, and Chytridiomycota, which together accounted for >95% of the fungal community and were clustered into one group (Appendix A). The first two dominant phyla and Chytridiomycota were not significantly different between stages, but the other two phyla, Mortierellomycota and Glomeromycota, differed at each of the depths (Figure 1C,D). Although three new phyla, Olpidiomycota, Calcarisporiellomycota, and Rozellomycota, were contained in the composition of the fungal community, the first two dominant phyla and significantly different phyla were unchanged at depths of 10–20 cm (Figure 1D). 

At the genus level, a total of 471 (44.60%) and 463 (44.01%) shared taxa at two depths were identified in the bacterial community across all stages, which was remarkedly higher than the unique taxa. However, only 54 (12.56%) and 49 (11.72%) shared taxa were identified in the fungal community, both of which were comparable with the unique taxa (Figure 2). Notably, variation in the unique taxa of the bacterial community was irregular and dramatically higher in A3 than in other years at depths of 0–10 cm (Figure 2A), while the unique taxa at depths of 10–20 cm decreased gradually from A0 to A20 and in subsequent years, increased slightly (Figure 2B). In contrast to the bacterial community, the unique taxa of the fungal community were the lowest in A3 at the two depths compared to the other stages, which was inconsistent with the results for the lowest total genus (Figure 2C,D). The differences in taxonomic composition and the unique taxa between stages potentially have significant effects on microbial community diversity.

### 3.2. Soil Microbial Community Diversity

Aerial sowing was found to have an enormous effect on microbial *α*- and *β*-diversity (Figure 3 and Figure 4). With the progress of afforestation stages, the richness indexes (Sobs, ACE and Chao1) of the bacterial community varied parabolically and reached the maximum at twenty years at both of depths, while those of the fungal community increased exponentially. The diversity indexes (Shannon, Simpson, and PD) of the bacterial community varied in accordance with richness, except for the Simpson index, which contrasted with the others (Figure 3). In comparison, the richness and diversity of the fungal community, except for the Simpson index, were remarkably lower than those of bacterial community. Overall, the richness indexes in A0 and A3 for both the bacterial and fungal communities were significantly lower than those in the other years at the two depths. At the phylum level, the bacterial community was clearly clustered into two groups, in which A0 and A3 were closer together than the other stages, and the variation of individual sites reduced with an increase in soil depth (Figure 4A,B). However, it was difficult to differentiate the constituents of the fungal community based on principal coordinate analysis and the Adonis test (Figure 4C,D). 

### 3.3. Determinants of Microbial Communities

Soil physicochemical properties have markedly divergent effects on the abundance of bacterial and fungal communities (Figure 5). For the bacterial community, the salt- (pH, EC, K, Mg, and Ca) and carbon-associated (TC and TOC) values were positively correlated with the abundance of Acidobacteria, Gemmatimonadota, Chloroflexi, and Planctomycetes but negatively correlated with the abundance of Actinobacteria, Bacteroidota, Firmicutes, and Patescibacteria (Figure 5A). This result was further confirmed by the redundancy analysis showing that pH, EC, K, Mg, Ca, SOC, and TC had significant contributions to the abundance of bacteria; these factors could explain 26.97% (RDA1 and RDA2) of the total variance (Figure 5B). For the fungal community, however, soil physicochemical properties were not significantly correlated with the first two dominant phyla, Ascomycota and Basidiomycota. Interestingly, SOC, EC, Ca, TC, and TN were positively associated with the abundance of Mortierellomycota and Mucoromycota (Figure 5C). Redundancy analysis showed that SOC, TC, EC, pH, and TP had significant contributions to the abundance of fungi; these soil properties could explain 18.12% of the total variance (Figure 5D). Mantel’s test showed that the *α*-diversity of bacterial communities was significantly correlated with TC, Ca, and EC; in addition to these three factors, fungal communities were also significantly correlated with SOC (Figure 6A). It was confirmed by the Random Forest that TC, Ca, and EC had significant contributions to the richness and diversity of the bacterial community; besides these three factors, TN, SOC, K, Mg, and AP also had significant contributions to the richness and diversity of the fungal community (Figure 6B). 

## 4. Discussion

### 4.1. Effects of Afforestation on Microbial Composition

Previously, a large number of synthesized studies showed that desert soil biomes are remarkedly distinct from other biomes in terms of their microbial composition [1,15]. Bacterial communities in the desert typically contain a number of ubiquitous phyla including Actinobacteria, Bacteroidetes, and Proteobacteria [1], which is inconsistent with our study (Figure 1). Unlike previous studies, we found that the second-most dominant phyla of the bacterial community in the study, Chloroflexi and Acidobacteria, accounted for a higher proportion than that in the hot desert and varied remarkably with stages of afforestation, as reported in the surrounding cold deserts [12,21,33]. Different from the bacterial community, afforestation by aerial sowing had fewer effects on the composition of the fungal community (Figure 1). The dominant phyla of the fungal community were also inconsistent with those reported in previous studies in the desert [15,34] and differed in both composition (Figure 1) and rare taxa (Figure 2). The dominant phyla, Ascomycota and Basidiomycota, accounted for a considerable proportion and varied insignificantly between stages of afforestation, which was inconsistent with the biological soil crusts of the surrounding site planted by shrubs in Tengger Desert [34]. This result may be attributable to a discrepancy in recovery time, which is more than 15 years for bacteria but ranges from decades to centuries for fungi. Both results indicate that the recovery of the fungal community is more difficult than that of the bacterial community over the short-term in a sandy desert environment. 

### 4.2. Effects of Afforestation on Microbial Diversity 

In addition to community composition, afforestation by aerial sowing also has great effects on the diversity of the soil microbial community, as reflected by the significant differences in the diversity of the bacterial community compared to the fungal community (Figure 3 and Figure 4). With the progress of afforestation stages, the difference in microbial community diversity vanished after 5 years when the sand dunes became fully fixed; this result is consistent with that of previous reports [12,17,21]. Cross-biome metagenomic analyses showed that desert microbial diversity, particularly for cold deserts, was actually far lower than that found in other biomes, and soil pH was found to be a reasonable predictor of prokaryotic diversity [5]. Based on this result, the phylotype richness under the current pH could range from 4500 to 6000, which is far higher than the richness of the bacterial community in the present study (Figure 3). A lower richness of the bacterial community was also observed in seriously saline–alkaline soil [12,21,35,36]. Conversely, soil with a relatively neutral pH was found to have greater bacterial richness (4000–4800) in the semi-arid Mu Us Sand Land, China [16]. Overall, the bacterial richness in the desert alkaline soil [12,21,37,38] was far less than that in the hot desert but comparable to that in the cold/polar desert [5,39]. This result may be attributable to the lower organic carbon input and higher soil salt in a harsh desert environment [37].

A high degree of dominance may contribute to lower diversity for the fungal community compared to the bacterial community (Figure 3), further limiting the formation of soil organic carbon (Appendix A) because soil fungi are more useful than bacteria in decomposing organic matter [40]. Although a few studies in semiarid deserts showed that the diversity of fungal communities was significantly affected by vegetation restoration duration [13,41], this result in inconsistent with our study (Figure 3). On the whole, the effects of afforestation on fungal diversity differed in terms of determinants and were particularly associated with soil carbon, nitrogen, and phosphorus availability (Figure 5 and Figure 6), which was the major restrictive factor for plant growth and nutrient acquisition among arbuscular mycorrhizal fungi in the desert [42,43]. These results confirm that soil microbial composition and diversity are strongly influenced by climate, vegetation type, and soil environmental factors.

### 4.3. Determinant of the Soil Microbial Community

Numerous studies showed that soil physicochemical properties and plant traits have a relatively dominant influence on the composition and diversity of soil microorganisms at a local scale [5,44] but divergent effects on bacterial and fungal communities [22,37,45]. For example, the composition of the bacterial community was governed primarily by soil physicochemical properties, whereas that of the fungal community was structured mainly by plant community composition during the restoration of grassland [22]. The soil physicochemical properties, including total carbon, pH, and total phosphorus, were found to be major abiotic factors driving the succession of soil bacterial communities during the restoration of the desert [12]. Given the low productivity and substantial unvegetated space in the desert [46,47], our results combined with others [5,39] suggest that the shape of the microbial community is regulated primarily by soil physicochemical properties.

Firstly, results indicate that soil pH can explain a large proportion of variation in the composition and diversity of the bacterial community at different spatial scales [18,19,20]. However, pH was not the key factor that determined difference between bacteria and fungi in this study (Figure 5 and Figure 6) as a result of the insignificant difference between stages (Appendix A) and the potential spillover effects of other chemical parameters [32]. Conversely, our findings showed that the soil physicochemical properties associated with salt (EC, K, Ca, and Mg) and carbon (TC and SOC) were the determinants of microbial diversity, particularly for the bacterial community (Figure 5 and Figure 6), which is in agreement with a previous study that emphasized the modulating role of salt and carbon inputs on the dynamics of soil microbes for saline land in the desert [12,35,48]. However, the positive relationship found between microbial diversity and soil-salt-associated properties disagrees with previous studies [48]. This result is likely associated with the microbial osmoregulation function [5] or attributable to the low salinity in this study (Appendix A), which was inadequate to restrain microbial development. Secondly, as one of the food sources for microorganisms, aboveground carbon input is essential to microbial metabolism and growth in vegetative succession; thus, it is not difficult to understand the positive effects of soil carbon on the abundance and diversity of microorganisms (Figure 5 and Figure 6). 

Moreover, studies showed that Actinobacteria and Proteobacteria play an important role in the soil carbon cycle by degrading recalcitrant carbon [22]. However, this function is seemingly restrained by low salt and carbon inputs, in which the abundance of Actinobacteria and Proteobacteria were negative and not correlated, respectively, with salt (EC, Mg and Ca) and carbon (TC and SOC) properties (Figure 5A). In contrast, the second-most dominant phyla, Chloroflexi and Acidobacteria, were closely associated with salt- and carbon-associated properties (Figure 5A), which suggests that the second-most dominant phyla of the bacterial communities play an important role in the soil carbon cycle. As a consequence, we speculate that the topsoil salt accumulation of xerophytes via salt secretion accelerated the development of the secondary dominant bacterial community (Figure 1) and thus increased soil carbon during the first 20 years (Appendix A). Compared to the bacterial community, the soil physicochemical properties were not related to the abundance of the dominant fungal community (Figure 5C) but did significantly contribute to the diversity of the fungal community (Figure 6). This result suggests that the soil’s physicochemical properties have positive effects on rare taxa with narrow niches, thus improving the diversity of the fungal community, which could explain why the dominant phyla of the fungal community were not significantly different between stages of afforestation (Figure 1C,D).

## 5. Conclusions

Similar to tree planting, large-scale afforestation by aerial sowing in the desert led to the near-natural recovery of degraded land and had great effects on the development of the soil microbial community, as reflected by the significant differences in composition and diversity between bare desert and afforestation land. The soil physicochemical properties were found to have markedly divergent effects on the phylum-specific abundance and diversity of bacterial and fungal communities, upon which salt and carbon inputs had greater effects than nutrient supply. Therefore, we conclude that afforestation through the topsoil salt and carbon inputs can promote the development of bacterial and fungal communities. Our findings provide new insights into the developmental processes and mechanisms of soil microbial communities in the desert.

## Figures and Tables

**Figure 1 jof-09-00399-f001:**
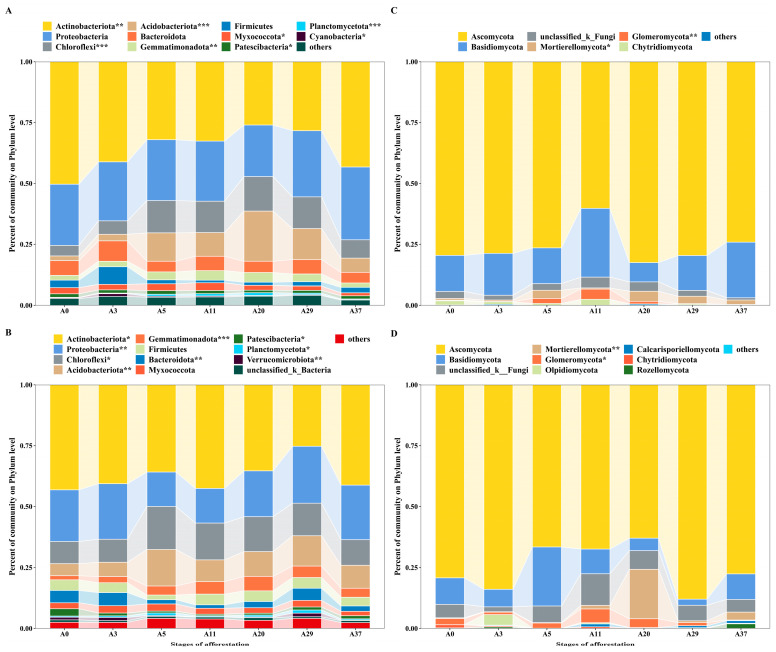
Community composition of bacteria ((**A**), 0–10 cm; (**B**), 10–20 cm) and fungi ((**C**), 0–10 cm; (**D**), 10–20 cm) at the phylum level for different stages of afforestation by aerial sowing in the northeastern Tengger Desert, China. The significance of the differences between stages was tested by a Kruskal–Wallis H test and represented as follows: * *p* < 0.05, ** *p* < 0.01, *** *p* < 0.001.

**Figure 2 jof-09-00399-f002:**
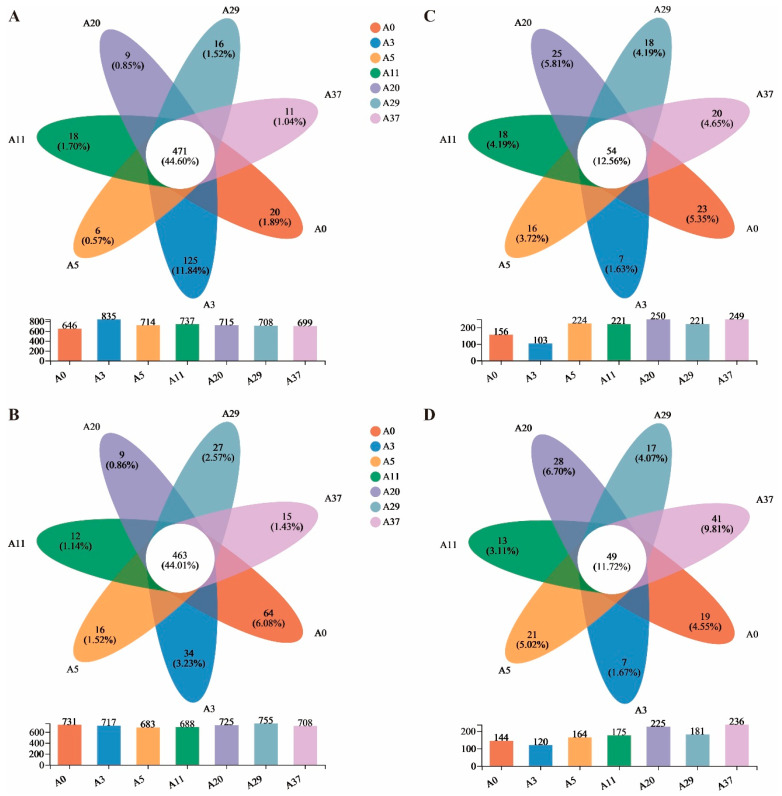
Venn diagram at OTU level of bacterial community ((**A**), 0–10 cm; (**B**), 10–20 cm) and fungal community ((**C**), 0–10 cm; (**D**), 10–20 cm) for different stages of afforestation by aerial sowing in the northeastern Tengger Desert, China.

**Figure 3 jof-09-00399-f003:**
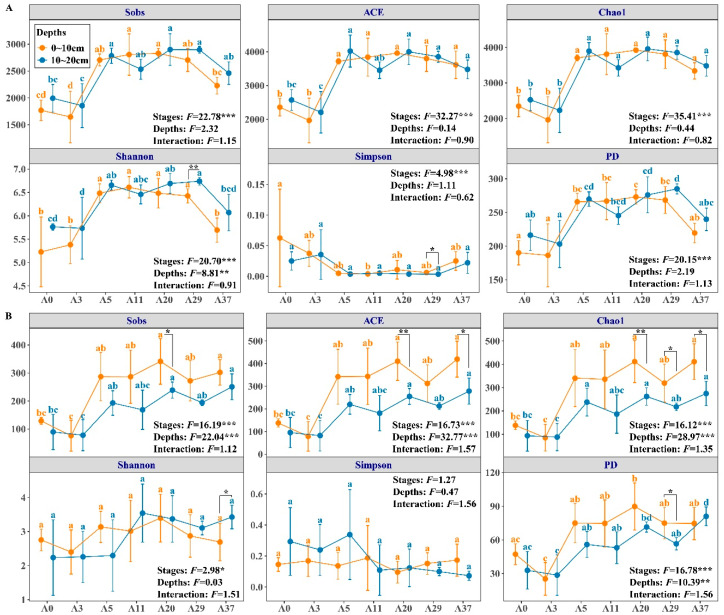
Changes in soil bacterial (**A**) and fungal (**B**) diversity indexes at the two depths during the stages of afforestation by aerial sowing in the northeastern Tengger Desert, China. The difference between the two depths was tested via ANOVA to show the significance level. The multiple comparisons of means between stages were tested using Tukey’s HSD and represented in lowercase. The multiple ANOVA of the interaction effect of stages, depths, and stages × depths on soil carbon is represented with the *F* value and significance level, * *p* < 0.05, ** *p* < 0.01, *** *p* < 0.001.

**Figure 4 jof-09-00399-f004:**
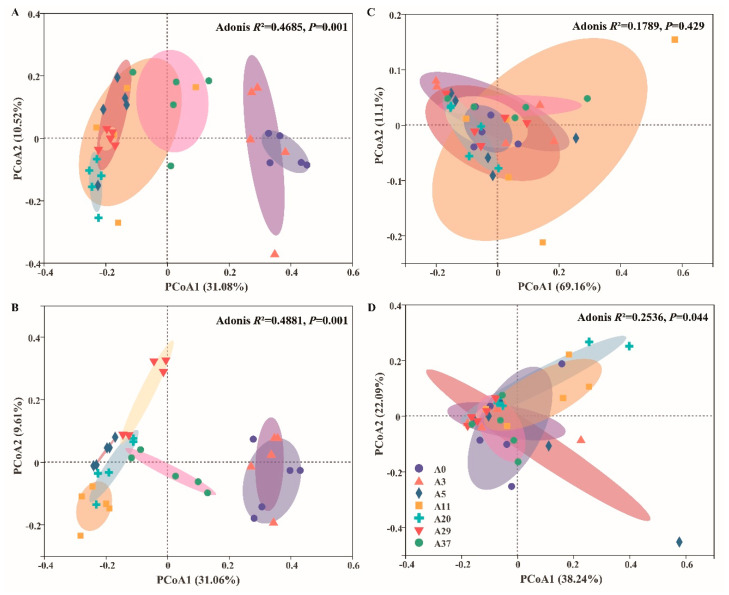
Principal coordinate analysis (PCoA) based on Bray–Curtis distances for differences between bacteria ((**A**), 0–10 cm; (**B**), 10–20 cm) and fungi ((**C**), 0–10 cm; (**D**), 10–20 cm) at the OTU level for different stages of afforestation by aerial sowing in the northeastern Tengger Desert, China. Adonis with 999 permutations was applied to test the effects of afforestation stages on community differences.

**Figure 5 jof-09-00399-f005:**
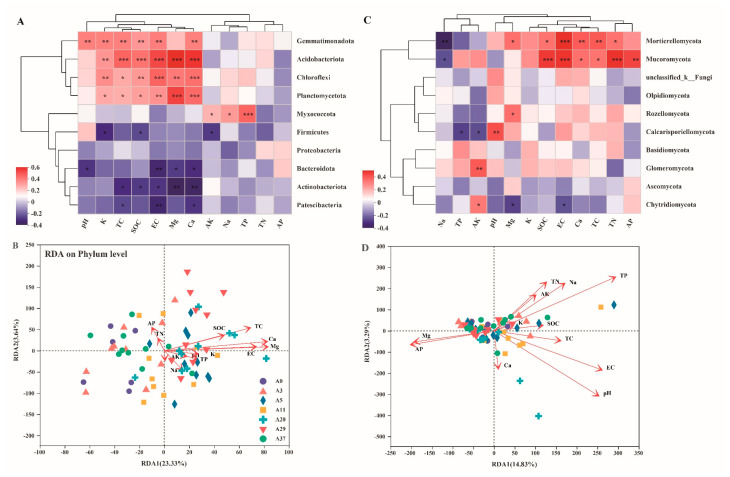
Spearman correlation heatmap and redundancy analysis (RDA) of the relationships between the dominant phyla (top 10) of bacteria (**A**,**B**) and fungi (**C**,**D**) and soil physicochemical properties filtered by variance inflation factor (VIF) for afforestation by aerial sowing in the northeastern Tengger Desert, China. Values on the x− and y−axes and the length of the corresponding arrows represent the importance of each soil physicochemical property in explaining the distribution of taxa across communities. Significance level, * *p* < 0.05, ** *p* < 0.01, *** *p* < 0.001.

**Figure 6 jof-09-00399-f006:**
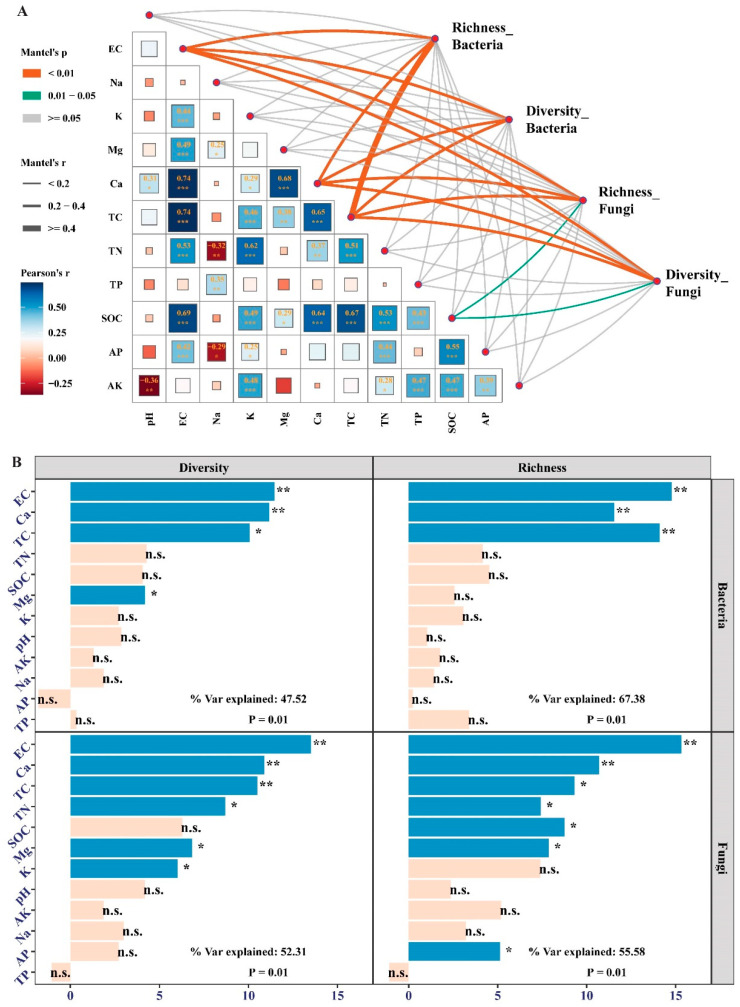
Pearson correlation and Mantel test between (**A**) the richness and diversity of bacterial and fungal communities and soil physicochemical properties and (**B**) the Random Forest calculated contribution of soil physicochemical properties to the richness and diversity of bacterial and fungal communities at different stages of afforestation by aerial sowing in the northeastern Tengger Desert, China. Significance level, * *p* < 0.05, ** *p* < 0.01, *** *p* < 0.001, and n.s., not significant.

## Data Availability

The metagenomic data have been submitted to the NCBI database with the Bio-Project accession number **PRJNA902373**.

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
