# Peer review of "Development and Determinants of Topsoil Bacterial and Fungal Communities of Afforestation by Aerial Sowing in Tengger Desert, China"

_jof, 2023, doi:10.3390/jof9040399_

Round 1
Reviewer 1 Report
In the manuscript data on a topical issue related to microbial succession in afforested deserts are presented. The authors found a number of important regularities between the environmental parameters and the structure of the developing bacterial and fungal communities. In the work, the authors used a comprehensive bioinformatics analysis to confirm their conclusions, which makes the work methodically competent.
Unfortunately, the quality of English does not make it possible to evaluate the results properly. I found various inaccuracies in terms, as well as grammatical errors and style issues. I strongly recommend authors to carefully check the text for grammar and style and/or give it to a native speaker for verification.
Author Response
In the manuscript data on a topical issue related to microbial succession in afforested deserts are presented. The authors found a number of important regularities between the environmental parameters and the structure of the developing bacterial and fungal communities. In the work, the authors used a comprehensive bioinformatics analysis to confirm their conclusions, which makes the work methodically competent.
Unfortunately, the quality of English does not make it possible to evaluate the results properly. I found various inaccuracies in terms, as well as grammatical errors and style issues. I strongly recommend authors to carefully check the text for grammar and style and/or give it to a native speaker for verification.
Response: Due to the limited time, we ask for a native speaker to check the quality of English. If it still not to meet the quality requirements, we will ask for a professional institution to do it.
Reviewer 2 Report
This research topic seems exciting and appropriate for publication in the Journal of Fungi (ISSN: 2309-608X). The manuscript, titled "Development and determinants of topsoil bacterial and fungal communities of afforestation by aerial seeding in desert, China" (jof-2285618), assessed the development and determinants of topsoil bacterial and fungal communities along nearly 40 years succession of afforestation by aerial seeding in Tengger Desert, China. Currently, the manuscript is in a rough condition. Major revisions are needed in two categories. The first is structural, and the second is technical. In order to meet minimum publication requirements, technical changes must be mandatory.
The manuscript must be revised prior to being submitted for fresh review.
Structural Changes:
Ensure that the reference style follows the journal format.
Technical Changes:
1. How did the authors differ from An et al. (2022) in their work? “Succession of soil bacterial community along a 46-year choronsequence artificial revegetation in an arid oasis-desert ecotone (https://doi.org/10.1016/j.scitotenv.2021.152496)”.
2. Better to mention Tengger Desert in the title as “Development and determinants of topsoil bacterial and fungal communities of afforestation by aerial seeding in Tengger Desert, China”.
3. Lines 21-23: This information needs to be rephrased as codes 5a, 0a, 3a, and 20a are given without any details, which may confuse the reader.
4. Lines 34-35: The information provided here is very basic and very common.
5. Lines 38-39: This information needs to be rephrased as code 20a are given without any details, which may confuse the reader.
6. Lines 46-36: Details regarding the deserts in China and their significance should be added after these lines.
7. 89-92: The hypothesis is missing, and the study aims are poorly described. The authors have divided the results section into three major sub-sections. It would be helpful if authors could incorporate those into the study aims by describing the clear points.
8. Lines 92-98: The following lines relate to materials and methods.
9. 2.3. DNA extraction and sequencing: This section needs to be rephrased, since the authors use the same words of expression as other researchers, which is an ethical violation. See more details:
“Colonization characteristics of fungi in Polygonum hydropipe L. and Polygonum lapathifolium L. and its effect on the content of active ingredients (https://doi.org/10.3389/fpls.2022.984483)”.
10. 2.4. Date processing: Is it date processing or data processing? Moreover, this section needs to be rephrased, since the authors use the same words of expression as other researchers, which is an ethical violation. See more details:
“Effects of Bellamya purificata Cultivation at Different Stocking Densities on the Dynamics and Assembly of Bacterial Communities in Sediment (https://doi.org/10.3390/biom13020254)”.
11. Figure 4. In my opinion, Figures 4A,B did not meet the requirements for using principal coordinates analysis (PCoA). The overall variation shown by PCoA1 and PCoA2 is less than 50%, which makes this method unsuitable for use.
12. Figure 5. In my opinion, Figures 5B,D did not meet the requirements for using redundancy analysis (RDA). The overall variation shown by RDAA1 and RDAA2 is less than 50%, which makes this method unsuitable for use.
13. Lines 316-416: Discussion. It would be more convenient to split 4.1. Effects of afforestation on microbial composition and diversity into 4.1. Soil microbial community composition and 4.2. Soil microbial community diversity. Rest rearranges the subsection order.
Author Response
This research topic seems exciting and appropriate for publication in the Journal of Fungi (ISSN: 2309-608X). The manuscript, titled "Development and determinants of topsoil bacterial and fungal communities of afforestation by aerial seeding in desert, China" (jof-2285618), assessed the development and determinants of topsoil bacterial and fungal communities along nearly 40 years succession of afforestation by aerial seeding in Tengger Desert, China. Currently, the manuscript is in a rough condition. Major revisions are needed in two categories. The first is structural, and the second is technical. In order to meet minimum publication requirements, technical changes must be mandatory.
The manuscript must be revised prior to being submitted for fresh review.
Structural Changes:
Ensure that the reference style follows the journal format.
Response: The reference was revised following the journal format.
Technical Changes:
- How did the authors differ from An et al. (2022) in their work? “Succession of soil bacterial community along a 46-year choronsequence artificial revegetation in an arid oasis-desert ecotone (https://doi.org/10.1016/j.scitotenv.2021.152496)”.
Response: There is three differences between our study and An et al. (2022): 1) the study area, our study was located at the northeast of Tengger Desert while An’s study located at the southwest of Badain Jaran Desert; 2) afforestation measures, our study sites were afforestation by aerial seeding while An’s study by tree planting, 3) object of microorganism, our study refer to bacterial and fungal communities, while An’s study only focus on bacterial community.
- Better to mention Tengger Desert in the title as “Development and determinants of topsoil bacterial and fungal communities of afforestation by aerial seeding in Tengger Desert, China”.
Response: Following the reviewer’s suggestion, the title was revised as “Development and determinants of topsoil bacterial and fungal communities of afforestation by aerial seeding in Tengger Desert, China”.
- Lines 21-23: This information needs to be rephrased as codes 5a, 0a, 3a, and 20a are given without any details, which may confuse the reader.
Response: The 5a, 0a, 3a, and 20a was revised as 5 yr, 0 yr, 3 yr and 20 yr.
- Lines 34-35: The information provided here is very basic and very common.
Response: Following the reviewer’s suggestion, the sentence was revised as ‘Afforestation has greater effects on composition and diversity of bacterial communities than fungal communities.’
- Lines 38-39: This information needs to be rephrased as code 20a are given without any details, which may confuse the reader.
Response: Following the reviewer’s suggestion, the 20a was revised as 20 yr.
- Lines 46-36: Details regarding the deserts in China and their significance should be added after these lines.
Response: Following the reviewer’s suggestion, the details regarding the deserts in China was added as ‘In China, the desert and the desertification-prone region are presently 4.21 × 106 km2, occupies 44% of China’s land area, that is a great challenge for combating desertification (Wang et al., 2022)’.
- 89-92: The hypothesis is missing, and the study aims are poorly described. The authors have divided the results section into three major sub-sections. It would be helpful if authors could incorporate those into the study aims by describing the clear points.
Response: Following the reviewer’s suggestion, the hypothesis was proposed as ‘1) composition of microbial community would change with plant community succession, 2) diversity of microbial community would increase with increased stage of afforestation, and 3) soil salt and nutrient status shape the composition and determine the diversity of microbial community’.
- Lines 92-98: The following lines relate to materials and methods.
Response: Following the reviewer’s suggestion, the sentence was removed and incorporate those into the materials and methods.
- 2.3. DNA extraction and sequencing: This section needs to be rephrased, since the authors use the same words of expression as other researchers, which is an ethical violation. See more details: “Colonization characteristics of fungi in Polygonum hydropipe L. and Polygonum lapathifolium L. and its effect on the content of active ingredients (https://doi.org/10.3389/fpls.2022.984483)”.
Response: Following the reviewer’s suggestion, this section was rephrased as follows: A 20 µL PCR reaction mixture consisted of 10 ng of template DNA and ddH2O, 4 μL 5 × Fast Pfu buffer, 2 μL 2.5 mM dNTPs, 0.8 μL each primer (5 μM), 0.4 μL Fast Pfu polymerase, and 0.2 μL BSA. PCR amplification cycling condition were designed following the Majorbio Bio-Pharm Technology Co., Ltd. (Shanghai, China). All samples were amplified in triplicate. Agarose gel with 2% concentration was used to extract the PCR product and the AxyPrep DNA Gel Extraction Kit (Axygen Biosciences, Union City, CA, USA) was employed to purify product. Purified amplicons were blended with equimolar amounts and paired-end sequenced at the Illumina MiSeq PE300 platform (Illumina, San Diego,USA).
- 2.4. Date processing: Is it date processing or data processing? Moreover, this section needs to be rephrased, since the authors use the same words of expression as other researchers, which is an ethical violation. See more details: “Effects of Bellamya purificata Cultivation at Different Stocking Densities on the Dynamics and Assembly of Bacterial Communities in Sediment (https://doi.org/10.3390/biom13020254)”.
Response: Data processing.
Response: Following the reviewer’s suggestion, this section was rephrased as follows: Raw sequencing data were de-multiplexed using an in-house perl script, and then quality-filtered by FASTP software (version 0.19.6) (Chen et al., 2018) and merged by FLASH software (version 1.2.7) (Magoc and Salzberg, 2011). The operational taxonomic units (OTU) was obtained through clustering the optimized sequences at 97% similarity level using UPARSE software (Edgar, 2013). A representative sequence for each OTU was selected as the most abundant sequence and then it was annotated and classified based on the RDP classifier Bayesian algorithm for bacteria and Unite database (http://unite.ut.ee/index.php) for fungi, respectively.
- Figure 4. In my opinion, Figures 4A, B did not meet the requirements for using principal coordinates analysis (PCoA). The overall variation shown by PCoA1 and PCoA2 is less than 50%, which makes this method unsuitable for use.
Response: Following the reviewer’s suggestion, we made an attempt to use the other method (e.g. PCA), we found that the PCA1 and PCA2 have less explanation of overall variation (36.16%, for bacteria at 0~10cm) than PCoA1 and PCoA2 (41.6%, Fig. 4A), and thus, we thought the PCoA was suitable for use in here, it just suggest the lower explanation of overall variation for bacteria but not for fungi (Fig. 4C).
- Figure 5. In my opinion, Figures 5B,D did not meet the requirements for using redundancy analysis (RDA). The overall variation shown by RDAA1 and RDAA2 is less than 50%, which makes this method unsuitable for use.
Response: Similarly, we made an attempt to use the other method (e.g. db-RDA), we found that the CAP1 and CAP2 have less explanation of overall variation (13.54%, for bacteria at 0~10cm) than RDA1 and RDA2 (26.97%, Fig. 5A), and thus, we thought the RDA was suitable for use in here, it just suggest the lower explanation of overall variation.
- Lines 316-416: Discussion. It would be more convenient to split 4.1. Effects of afforestation on microbial composition and diversity into 4.1. Soil microbial community composition and 4.2. Soil microbial community diversity. Rest rearranges the subsection order.
Response: Following the reviewer’s suggestion, this section was rearranged into two parts.
Round 2
Reviewer 1 Report
The new version looks much better in relation to English.
Other corrections are not required
Reviewer 2 Report
I reviewed this manuscript in the first round and provided detailed comments for improvement, including structural and technical changes. In making structural changes, the author was instructed to follow the journal's reference style instructions. Although the authors stated that they followed the reference style, they did not. They may need to view the latest articles published in the journal in order to understand the correct reference style. Furthermore, there is a lack of track changes (or color content) information which indicates what changes were made by the author. The reviewer cannot view the revised version without those changes. Authors are responsible for providing such information.